# AN EQUAL-SIZE HARD EM ALGORITHM
# FOR DIVERSE DIALOGUE GENERATION

**Yuqiao Wen**[1]   **Yongchang Hao**[1]   **Yanshuai Cao**[2]   **Lili Mou**[1,3]
[1]Dept. Computing Science, Alberta Machine Intelligence Institute (Amii), University of Alberta
[2]Borealis AI   [3]Canada CIFAR AI Chair, Amii
`yq.when@gmail.com`  `yongcha1@ualberta.ca`
`yanshuai.cao@borealisai.com`  `doublepower.mou@gmail.com`

## ABSTRACT

Open-domain dialogue systems aim to interact with humans through natural language texts in an open-ended fashion. Despite the recent success of super large dialogue systems such as ChatGPT, using medium-to-small-sized dialogue systems remains the common practice as they are more lightweight and accessible; however, generating diverse dialogue responses is challenging, especially with smaller models. In this work, we propose an Equal-size Hard Expectation–Maximization (EqHard-EM) algorithm to train a multi-decoder model for diverse dialogue generation. Our algorithm assigns a sample to a decoder in a hard manner and additionally imposes an equal-assignment constraint to ensure that all decoders are well-trained. We provide detailed theoretical analysis to justify our approach. Further, experiments on two large-scale open-domain dialogue datasets verify that our EqHard-EM algorithm generates high-quality diverse responses[1].

## 1 INTRODUCTION

Open-domain dialogue systems aim to generate natural language text utterances to hold open-ended conversations with humans (Li et al., 2017a; Wang et al., 2021b). These systems have shown great success, and are seamlessly integrated into our society through chatbots. The recently launched ChatGPT[2] model, for example, has shown remarkable conversational skills. However, ChatGPT is prohibitively large in size and requires significant human feedback during its training process, and therefore, training medium-to-small-sized language models without human feedback still remains the common practice (Wu et al., 2021; Wang et al., 2021a; Chen et al., 2022), and these smaller models tend to generate generic responses such as *I don't know* (Li et al., 2016b; Wang et al., 2021b). One of the possible causes is the *one-to-many mapping* phenomenon in the dialogue task, where a dialogue context may correspond to multiple plausible responses (Wei et al., 2019; Bao et al., 2020; Khan et al., 2020). Learning to generate a dialogue utterance is thus analogous to learning a *multimodal* distribution of a continuous variable, where a *mode* refers to a peak in the distribution; in the dialogue task, a mode can be thought of as a set of similar responses. The widely used cross-entropy training is not good at capturing different modes, as it encourages the prediction to cover all plausible responses, forcing the model to learn an overly smooth distribution. Consequently, these neural dialogue models resort to generating generic responses (Wei et al., 2019).

Previous studies address generic responses mainly at training or inference time. Training-time approaches explore different training objectives: Li et al. (2016b) apply reinforcement learning to optimize a customized reward function that discourages generic responses. Khan et al. (2020) train a generative adversarial network in the latent space to discourage generic responses because they can be easily identified by the discriminator. Among inference approaches, Vijayakumar et al. (2016) encourage diverse beam results by penalizing similar beam entities. Holtzman et al. (2020) apply nucleus sampling to allow plausible but less probable words to be decoded. Wang et al. (2021b) apply label smoothing to prevent the model from being overly confident with generic responses.

---

[1]Our code is available at `https://github.com/MANGA-UOFA/EqHard-EM`
[2]`https://openai.com/blog/chatgpt/`

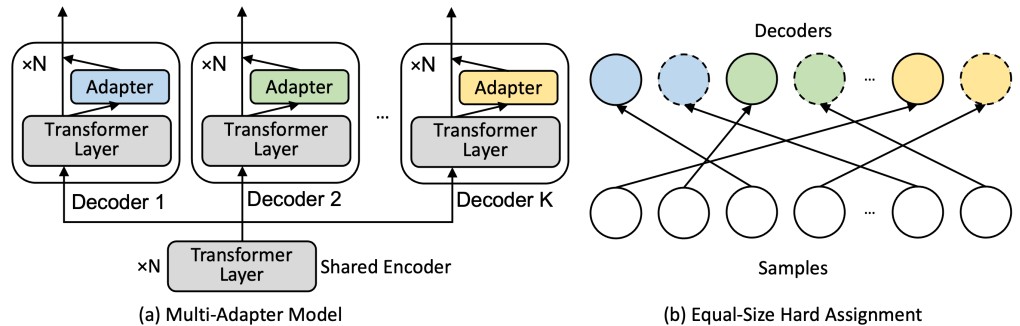

Figure 1: (a) Our multi-adapter neural architecture. (b) The equal-size hard assignment scheme. Dashed circles: decoders are conceptually duplicated when we solve the assignment problem.

Another way to address the generic responses is to explicitly capture different dialogue patterns with mixture models. This has been explored for diverse machine translation, where Shen et al. (2019) train a multi-decoder model with the Expectation–Maximization (EM) algorithm (Dempster et al., 1977) to address the one-to-many mapping phenomenon. However, their direct application of the standard EM algorithms suffers from the *decoder-collapsing problem*, where the multi-decoder model degenerates to a single-decoder model. The authors attempt to alleviate this problem by disabling the dropout layers during the E-step of the EM algorithm, but their results show that the model is still susceptible to collapses.

To this end, we propose a novel EM variant to multi-decoder diverse dialogue generation. A standard EM algorithm assigns a sample to all decoders by the posterior probability, thus known as Soft-EM; it suffers from *synchronous-training* collapse, where the posterior probabilities tend to be similar and all decoders are also trained similarly. The Hard-EM variant trains the decoder that has the highest posterior probability; it suffers from *non-training* collapse due to the rich-gets-richer phenomenon. Our proposed EM algorithm avoids both types of collapses by adopting hard assignments and imposing an equal-assignment constraint. Hence, we call our approach Equal-size Hard EM (EqHard-EM).

We conducted experiments on the Weibo (Gao et al., 2019) and OpenSubtitles (Lison et al., 2018) datasets. Results show that our EqHard-EM algorithm alleviates the decoder-collapsing problem in both Soft-EM and Hard-EM, allowing the multi-decoder model to specialize and generate high-quality and diverse responses. In addition, we provide in-depth theoretical and empirical analyses to better understand our EqHard-EM algorithm.

To sum up, our contributions are three-fold: 1) We propose the EqHard-EM algorithm to alleviate the collapse issues in soft and hard EM variants; 2) We provide detailed theoretical analysis to justify our equal assignment; and 3) We conduct extensive empirical experiments to show the effectiveness of our approach.

## 2 APPROACH

In this section, we first present our mixture model architecture (Subsection 2.1). Then, we propose EqHard-EM, a novel EM variant, for training such mixture models (Subsection 2.2).

### 2.1 NEURAL ARCHITECTURE

We propose to address the one-to-many mapping phenomenon in the dialogue task with a mixture model. Given a dialogue context, we use a shared encoder to build an input utterance's hidden representations, based on which multiple decoders generate a set of output responses.

For multiple decoders, it is possible to instantiate each with a full Transformer model (Shen et al., 2019), but this leads to a large number of parameters that may not fit into the memory of an ordinary GPU. To this end, we propose a multi-adapter architecture (Figure 1a), where different decoders share most Transformer parameters, but differ by a few inserted, thin adapter layers (Houlsby et al., 2019). In fact, the adapter model is widely used for parameter-efficient domain adaptation (Artetxe et al., 2020; Wang et al., 2021a; Ngo Trung et al., 2021). We propose to apply this architecture to

mixture models. Our preliminary experiments show that the multi-adapter model results in a degradation of one BLEU point, but reduces 76% parameters compared with full-Transformer decoders. Therefore, we use the multi-adapter model in our main experiments.

Specifically, an adapter layer applies two projections of linear transformation and non-linear activation. Further, a residual connection is included to ensure that the Transformer layers are not significantly disturbed. This process can be formulated as

$$\text{AdapterLayer}(\mathbf{x}) = \mathbf{x} + \mathbf{W}_1(\phi(\mathbf{W}_2\mathbf{x})) \tag{1}$$

where $\mathbf{x} \in \mathbb{R}^d$ is the input to the adapter layer; $\mathbf{W}_1 \in \mathbb{R}^{d \times d'}$ and $\mathbf{W}_2 \in \mathbb{R}^{d' \times d}$ are the projection matrices with $d' < d$; and $\phi$ is a non-linear activation function such as the rectified linear unit (ReLU, Agarap, 2018).

## 2.2 TRAINING METHODS

We propose to train our multi-adapter architecture in two stages: 1) Pretraining of the shared Transformer-layer parameters, and 2) EM training of adapter-layer parameters as the mixture model. In the first stage, we pretrain a T5-small[3] encoder–decoder model (Raffel et al., 2020) with a standard cross-entropy loss since the Transformer parameters are shared among different mixture components; in the second stage, we adopt the standard treatment of adapter modules (Houlsby et al., 2019) and freeze the shared Transformer parameters to minimize the number of trainable parameters for effective adaptation.

In the rest of this subsection, we focus on the second stage, i.e., training the adapter parameters of the mixture model. We first review classic EM algorithms and analyze their drawbacks. Then, we propose a novel EM variant that addresses the drawbacks.

**Classic EM Algorithms.** Consider a multi-decoder model with $K$ decoders. Given a dialogue context $\mathbf{c}$ and response $\mathbf{r}$, our training objective is to maximize the marginal log-likelihood $\log \sum_k P(\mathbf{r}, z = k|\mathbf{c}; \boldsymbol{\theta})$, where $\boldsymbol{\theta}$ is the model parameters and the variable $z \in \{1, \cdots, K\}$ is the choice of the decoder. Since $z$ is unobserved in the training set, it can be viewed as a latent variable, and we apply the expectation–maximization (EM) framework (Dempster et al., 1977) as a principled approach to training latent-variable models. Specifically, the EM algorithm alternates between the E-step and M-step:

• *E-step*: Estimate the posterior distribution of the latent variable by applying Bayes' rule:

$$Q_k(\mathbf{c}, \mathbf{r}) = P(z = k|\mathbf{c}, \mathbf{r}; \boldsymbol{\theta}) = \frac{P(\mathbf{r}|z = k, \mathbf{c}; \boldsymbol{\theta})P(z = k|\mathbf{c}; \boldsymbol{\theta})}{\sum_{k'=1}^{K} P(\mathbf{r}|z = k', \mathbf{c}; \boldsymbol{\theta})P(z = k'|\mathbf{c}; \boldsymbol{\theta})} \tag{2}$$

• *M-step:* Maximize the expected log-likelihood with respect to $\boldsymbol{\theta}$:

$$\mathbb{E}_{z \sim Q_k(\mathbf{c}, \mathbf{r})}[\log P(\mathbf{r}, z|\mathbf{c}; \boldsymbol{\theta})] = \sum_{k=1}^{K} Q_k(\mathbf{c}, \mathbf{r}) \log P(\mathbf{r}, z|\mathbf{c}; \boldsymbol{\theta}) \tag{3}$$

We additionally assume a uniform prior for every context $\mathbf{c}$ as $P(z = k|\mathbf{c}; \boldsymbol{\theta}) = 1/K$, following the one-to-many assumption that each context has multiple plausible responses. This assumption simplifies Eqn. (2) as

$$P(z = k|\mathbf{c}, \mathbf{r}; \boldsymbol{\theta}) = \frac{P(\mathbf{r}|z = k, \mathbf{c}; \boldsymbol{\theta})}{\sum_{k'=1}^{K} P(\mathbf{r}|z = k', \mathbf{c}; \boldsymbol{\theta})} \tag{4}$$

suggesting that the posterior is proportional to the likelihood under the uniform prior assumption.

• *Soft-EM.* The classic EM algorithm, as explained above, assigns a sample to a mixture component by the posterior distribution (2) in a soft manner, and thus is also known as Soft-EM. However, we

---

[3]In principle, other pretrained language models may also be used. In our development, we have tried finetuning GPT-2 on the OpenSubtitles dataset. It gave a similar, but slightly lower, result: 3.26 BLEU2-F vs T5-small's 3.46 BLEU2-F. Therefore, we chose T5-small as the base model, because it achieves higher performance with fewer parameters.

observe that Soft-EM performs poorly in our experiments due to **synchronous-training collapse**[4], where all decoders are trained almost synchronously and generate similar responses. In particular, the Transformer-layer parameters are pretrained. Even with the inserted adapter layers, all decoders still behave similarly, yielding a similar posterior distribution (2). This in turn yields a similar gradient for each decoder, and as a result, multiple decoders are virtually collapsed to one decoder.

• *Hard-EM.* Alternatively, the assignment in the E-step can be accomplished in a hard manner by choosing the best-fit mixture component

$$k^* = \mathrm{argmax}_k \, P(z = k|\mathbf{c}, \mathbf{r}; \boldsymbol{\theta}) \tag{5}$$

Then, the M-step is approximated as follows

$$\mathbb{E}_{z \sim Q_k(z|\mathbf{c}, \mathbf{r}; \boldsymbol{\theta})}[\log P(\mathbf{r}, z|\mathbf{c}; \boldsymbol{\theta})] \approx \log P(\mathbf{r}, z = k^*|\mathbf{c}; \boldsymbol{\theta}) \tag{6}$$

Unfortunately, the Hard-EM algorithm suffers from the **non-training collapse**, where only one decoder is trained due to the rich-gets-richer phenomenon. That is to say, a particular decoder will be selected by Eqn. (5) due to the random initialization of adapter layers. Then, that decoder will be trained with more samples and perform better than the other decoders, making it even more likely to be selected by Eqn. (5). In practice, we observe that only one decoder is properly trained by the Hard-EM algorithm, making the mixture model ineffective.

**Our proposed EqHard-EM**. To this end, we propose an Equal-size Hard Expectation–Maximization (EqHard-EM) algorithm to address the collapse issues. Specifically, we adopt the hard assignments from Hard-EM to avoid synchronous-training collapse, whereas we impose an equal-assignment constraint to ensure that all decoders are adequately trained to avoid non-training collapse.

We formulate the E-step as a constrained optimization problem over an assignment matrix $\mathbf{A} \in \{0, 1\}^{N \times K}$, where $N$ is the number of samples and $K$ is the number of decoders. The element $A_{nk}$ indicates whether the $n$th sample is assigned to the $k$th decoder. We define the cost matrix $\mathbf{C} \in \mathbb{R}^{N \times K}$ by the posterior probability, i.e., $C_{nk} = -Q_k(\mathbf{r}^{(n)}, \mathbf{c}^{(n)})$ as in Eqn. (2). Our equal-size hard E-step aims to

$$\mathrm{minimize}_{\mathbf{A}} \quad \sum_{k=1}^{K} \sum_{n=1}^{N} A_{nk} C_{nk} \tag{7}$$

$$\mathrm{subject\ to} \quad \sum_{k=1}^{K} A_{nk} = 1, \quad \mathrm{for}\ n = 1, \cdots, N \tag{8}$$

$$\sum_{n=1}^{N} A_{nk} = N/K, \quad \mathrm{for}\ k = 1, \cdots, K \tag{9}$$

$$A_{nk} \in \{0, 1\} \tag{10}$$

Compared with Hard-EM, we impose the constraint (9) to ensure that every decoder is assigned an equal number of samples.

In practice, this optimization problem can be transformed into a balanced assignment problem, as shown by Figure 1b. This is accomplished by duplicating each decoder and its associated costs by a factor of $\frac{N}{K}$ to match the number of samples. Here, we enforce the batch size $N$ to be divisible by $K$. Note that the decoders are only conceptually duplicated, and it does not incur additional computational cost. We solve the balanced assignment problem by the classic Hungarian algorithm (Kuhn, 1955), which was later improved by Jonker & Volgenant (1987) to achieve a computational complexity of $O(N^3)$. This is efficient in our batch implementation and contributes to less than 1% of the total training time.

For inference, we follow existing literature (Zhao et al., 2017; Gu et al., 2019; Shen et al., 2019) and consider the multi-response generation setting, where the goal is to generate a set of diverse responses given a dialogue context. We collect the outputs of different decoders by greedy decoding.

**Theoretical analysis.** We provide theoretical justifications for our EqHard-EM algorithm. Our equal-assignment constraint makes sense intuitively, because the aggregated posterior should be the same as the prior distribution, suppose the posterior distribution is correctly estimated and we have

---

[4]Shen et al. (2019) refer to this phenomenon as the "D2 Degeneracy" and interpret it as the bypassing of latent variables, similar to the posterior collapse of variational auto-encoders (VAE, Bowman et al., 2016; Bahuleyan et al., 2018). We find the connection with VAE weak. Instead, we present our own interpretation of synchronous training, which is justified by empirical evidence.

enough samples. This is given by $\sum_{\mathbf{c}} \sum_{\mathbf{r}} P(z|\mathbf{r},\mathbf{c})P(\mathbf{r},\mathbf{c}) = \sum_{\mathbf{c}} P(z|\mathbf{c})P(\mathbf{c}) = \sum_{\mathbf{c}}(1/K)P(\mathbf{c}) = 1/K$ for $z = 1,\cdots,K$. In this part, we develop a theorem on how equal the assignments would be in the imperfect-posterior and finite-sample cases.

**Theorem 1.** *With probability at least* $1-\delta$*, we have*

$$\left| \frac{1}{|\mathcal{D}|} \sum_{(\mathbf{r},\mathbf{c}) \in \mathcal{D}} Q(z|\mathbf{r},\mathbf{c}) - \frac{1}{K} \right| \le \sqrt{\frac{1}{2|\mathcal{D}|} \log\left(\frac{2K}{\delta}\right)} + \mathbb{E}_{\mathbf{c},\mathbf{r}}\left[ \left| Q(z|\mathbf{c},\mathbf{r}) - p(z|\mathbf{c},\mathbf{r}) \right| \right]$$

*for all* $z \in \{1,\cdots,K\}$*. Here,* $Q(z|\mathbf{c},\mathbf{r})$ *is an estimated posterior and* $\mathcal{D}$ *is a dataset with finite samples.*

The proof sketch is to consider finite samples (Lemma 1) and the estimated posterior (Lemma 2) separately, which can then be combined in a straightforward manner. See Appendix A for detailed proof.

Theorem 1 justifies the equal-assignment constraint, because with a high probability, the assignments will converge to uniform if we have a large dataset and a near-correct posterior. Note that the term $\sum_{(\mathbf{r},\mathbf{c}) \in \mathcal{D}} Q(z|\mathbf{r},\mathbf{c})$ in the theorem mainly works for soft assignments. However, we could not adopt soft assignment in the algorithm because it leads to synchronous-training collapse. The theorem is nevertheless illuminating to our algorithm, which is a hard-assignment approximation similar to Hard-EM.

## 3 EXPERIMENTS

### 3.1 SETUP

**Datasets.** We evaluated our approach on two large-scale open-domain dialogue datasets: Weibo (Gao et al., 2019) and OpenSubtitles (Lison et al., 2018). The Weibo dataset is a compilation of posts and replies, crawled from a Chinese micro-blogging website. The dataset features explicit one-to-many mapping because in most cases, a post corresponds to multiple replies. On the other hand, the OpenSubtitles dataset is a collection of movie dialogues, which are parsed from online subtitle files. The dataset is known to be noisy (Vinyals & Le, 2015; Feng et al., 2020; Nakazawa et al., 2021), and therefore it demonstrates the implicit one-to-many phenomenon: a movie utterance may lead to many plausible responses even though the dataset only contains one reference response. Admittedly, there exist other manually constructed, high-quality datasets such as DailyDialog (Li et al., 2017b), but they are usually much smaller and the one-to-many phenomenon is less severe.

Recently, Wen et al. (2022) identify significant overlapping between training, validation, and/or test splits in various open-domain dialogue datasets (including OpenSubtitles) due to the oversights of data collection. We use their cleaned OpenSubtitles dataset, which contains 1M, 12K, and 12K samples for training, validation, and test, respectively. We investigated the Weibo dataset and found significant overlapping between the validation and test sets as well, so we followed their cleaning strategy and obtained 4M, 10K, and 10K training, validation, and test samples. Overall, Weibo and OpenSubtitles allow us to conduct comprehensive analyses in both explicit and implicit one-to-many mapping scenarios and in different languages.

**Metrics.** Open-domain dialogue evaluation is currently an active area of research due to its one-to-many nature (Tao et al., 2018; Sinha et al., 2020). In particular, the standard BLEU metric, which measures the $n$-gram similarity between generated and reference responses, has a low correlation with human evaluation because a plausible response is not necessarily similar to the reference response (Liu et al., 2016). Therefore, we followed previous work on open-domain dialogue (Zhao et al., 2017; Gu et al., 2019; Li et al., 2020) and generated 10 responses for each dialogue context to compute BLEU-P, BLEU-R, and BLEU-F. Roughly, BLEU-P measures the average quality of the generated responses, whereas BLEU-R measures how well the references are covered. BLEU-F is the harmonic mean.[5] We adopted uni-gram and bi-gram BLEU scores in our experiments, because Liu et al. (2016) show that the $n$-gram overlap is often zero for dialogue evaluation with $n > 2$.

---

[5]The terminologies of BLEU-P, BLEU-R, and BLEU-F are inspired by, but different from, the precision and recall in information retrieval (Manning et al., 2008). Interested readers are referred to Zhao et al. (2017) for the details of BLEU-P, BLEU-R, and BLEU-F.

| Dataset | Model | BLEU1$^\uparrow$ | | | BLEU2$^\uparrow$ | | | Distinct $n$-gram$^\uparrow$ | | Pairwise |
|---|---|---|---|---|---|---|---|---|---|---|
| | | F | P | R | F | P | R | Dist-1 | Dist-2 | BLEU$^\downarrow$ |
| Weibo | T5-small beam search | 18.59 | 25.59 | 14.60 | 8.51 | **17.57** | 5.62 | 10.41 | 14.17 | 65.46 |
| | T5-small nucleus sampling$^\dagger$ | 20.76 | 26.13 | 17.23 | 10.03 | 17.44 | 7.04 | 16.90 | 23.79 | 44.55 |
| | AdaLabel (Wang et al., 2021b) | 14.80 | 21.63 | 11.25 | 5.86 | 13.72 | 3.72 | 10.41 | 14.71 | 62.46 |
| | DialogBERT (Gu et al., 2021)$^\dagger$ | 15.99 | 15.88 | 16.10 | 3.55 | 3.95 | 3.23 | 68.95 | 96.76 | 0.01 |
| | Our approach | **22.65** | **26.17** | **19.96** | **10.13** | 14.55 | **7.77** | 27.03 | 42.02 | 22.32 |
| Open-Subtitles | T5-small | 14.39 | 11.23 | 20.01 | 3.46 | **2.40** | 6.22 | 13.38 | 20.71 | 42.76 |
| | T5-small nucleus sampling$^\dagger$ | 16.03 | 11.21 | 28.13 | 2.51 | 1.47 | 8.48 | 60.50 | 91.26 | 0.29 |
| | AdaLabel (Wang et al., 2021b) | 15.80 | 11.94 | 23.38 | 3.20 | 2.19 | 5.94 | 26.47 | 43.57 | 17.21 |
| | DialogBERT (Gu et al., 2021)$^\dagger$ | 12.86 | 8.71 | 24.50 | 1.20 | 0.68 | 4.94 | 69.70 | 96.49 | 0.02 |
| | Our approach | **17.86** | **12.98** | **28.60** | **3.76** | 2.36 | **9.27** | 34.09 | 50.41 | 13.15 |

Table 1: Results on Weibo and OpenSubtitles. We use cleaned versions of these datasets since the original datasets contain overlapping samples (Wen et al., 2022). $^\dagger$ indicates sampling-based approaches; DialogBERT performs poorly with beam search or greedy decoding, also reported by Wen et al. (2022). $^{\uparrow/\downarrow}$The higher/lower, the better. Underlined numbers are the best diversity scores (Dist and Pairwise-BLEU) among non-sampling methods. Note that comparing diversity is only meaningful when BLEU scores are controlled.

Consequently, high-order BLEU scores mostly rely on smoothing techniques and correlate poorly with human satisfaction.

For diversity, we adopt the Dist (Li et al., 2016a) and Pairwise-BLEU (Shen et al., 2019) scores. Dist-$n$ is a widely used metric that measures the percentage of distinct $n$-grams among utterances. We follow the standard practice and report Dist-1 and Dist-2 between the generated responses for each dialogue context (Li et al., 2016a; Wang et al., 2021b). In addition, we follow Shen et al. (2019) and report Pairwise-BLEU, which measures the pairwise similarity among the responses, i.e., a low Pairwise-BLEU indicates high diversity.

**Implementation details.** We considered 10 decoders due to the setup of diverse generation evaluation (Zhao et al., 2017; Gu et al., 2019; Li et al., 2020). Each E-step involved 640 samples, assigning 64 samples to each decoder. More implementation details are presented in Appendix B.

## 3.2 RESULTS AND ANALYSES

**Main results.** Table 1 shows our main results on Weibo and OpenSubtitles. The T5-small model is a pretrained encoder–decoder Transformer (Raffel et al., 2020), where we experimented with both beam search and sampling. For beam search, we set the beam size to be 10, i.e., the number of samples needed for evaluation. As known, beam search does not generate diverse responses (Vijayakumar et al., 2016; Holtzman et al., 2020), shown by low BLEU-R and Dist scores, but high Pairwise-BLEU. The performance of nucleus sampling is inconsistent in the two experiments: on the Weibo dataset, it slightly improves both generation quality and diversity; on OpenSubtitles, however, the generated responses are much more diverse, but the generation quality is not convincingly improved, as shown by the low BLEU2-F score. We examined the samples and believe this is because nucleus sampling depends on the distribution temperature and the nucleus size for sampling: if the sampling scope is too small, then the diversity would not be improved much; if otherwise the sampling scope is too large, the quality may deteriorate.

By contrast, our EqHard-EM approach is able to perform consistently well on both datasets by simple greedy decoding. Our BLEU-R is much higher than most competing methods. This shows that our EqHard-EM has a high coverage of potential responses, largely alleviating the one-to-many issue in dialogue generation. Overall, we achieve the highest BLEU-F scores, while largely improving diversity. In fact, we outperform all non-sampling methods in terms of both Dist and Pairwise-BLEU scores.

We also experimented with recently proposed models: AdaLabel (Wang et al., 2021b) is an encoder–decoder Transformer, trained with adaptive label smoothing to improve diversity. DialogBERT (Gu et al., 2021) features the use of a BERT encoder (Devlin et al., 2019) to capture contextual information. They were alleged to be state-of-the-art models when the dataset-overlapping problem was not realized. However, they perform poorly on the deduplicated datasets; our results in the diverse generation setting are consistent with single-response generation (Wen et al., 2022).

| Dataset | Model | BLEU1↑ | | | BLEU2↑ | | | Distinct $n$-gram↑ | | Pairwise BLEU↓ |
|---------|-------|------|------|------|------|------|------|--------|--------|------|
| | | F | P | R | F | P | R | Dist-1 | Dist-2 | |
| Weibo | Beam search | 18.59 | 25.59 | 14.60 | 8.51 | **17.57** | 5.62 | 10.41 | 14.17 | 65.46 |
| | Soft-EM | 17.02 | **27.55** | 12.32 | 6.27 | 16.03 | 3.89 | 10.69 | 12.45 | 86.25 |
| | Hard-EM | 4.81 | 3.09 | 10.84 | 2.15 | 1.61 | 3.22 | 4.01 | 4.11 | 0.00 |
| | EqRandom-Dynamic | 18.72 | 27.34 | 14.23 | 7.21 | 15.87 | 4.66 | 12.79 | 16.30 | 71.19 |
| | EqRandom-Fixed | 20.01 | 25.51 | 16.46 | 8.12 | 14.41 | 5.65 | 19.40 | 25.77 | 46.60 |
| | EqHard-EM | **22.65** | 26.17 | **19.96** | **10.13** | 14.55 | **7.77** | 27.03 | 42.02 | 22.32 |
| OpenSubtitles | Beam search | 14.39 | 11.23 | 20.01 | 3.46 | 2.40 | 6.22 | 13.38 | 20.71 | 42.76 |
| | Soft-EM | 15.48 | 14.55 | 16.54 | 3.03 | **2.76** | 3.35 | 12.89 | 15.35 | 72.95 |
| | Hard-EM | 6.84 | 4.13 | 19.81 | 1.25 | 0.73 | 4.30 | 4.83 | 4.81 | 1.50 |
| | EqRandom-Dynamic | 15.80 | 14.56 | 17.27 | 2.92 | 2.56 | 3.40 | 14.52 | 18.57 | 70.12 |
| | EqRandom-Fixed | 15.89 | **14.59** | 17.46 | 2.93 | 2.56 | 3.43 | 14.73 | 18.96 | 68.89 |
| | EqHard-EM | **17.86** | 12.98 | **28.60** | **3.76** | 2.36 | **9.27** | 34.09 | 50.41 | 13.15 |

Table 2: Comparing EM-like algorithms based on multi-adapter T5-small. The Soft-EM and Hard-EM variants follow Shen et al. (2019), who address diverse machine translation.

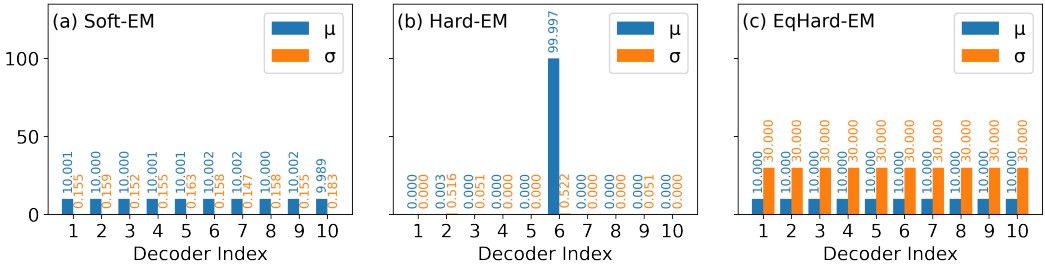

Figure 2: Decoder assignment statistics for (a) Soft-EM, (b) Hard-EM, and (c) EqHard-EM. Means and standard deviations are computed on the Weibo training set.

**Comparing EM variants.** We conducted controlled experiments to compare EM-like algorithms (Table 2). We find Soft-EM suffers from synchronous-training collapse, as it achieves very high Pairwise-BLEU scores. It achieves even lower diversity than beam search because of the synchronous-training collapse. This can be best seen by decoder assignment statistics (Figure 2a), where every decoder is assigned an equal fraction of samples with near-zero standard deviation.

On the other hand, the Hard-EM algorithm also performs poorly. It suffers from the non-training collapse, due to the rich-gets-richer phenomenon. As seen in Figure 2b, only Decoder 6 is trained in the experiment. Hard-EM achieves near-zero Pairwise-BLEU and the lowest Dist scores (Table 2) because the untrained decoders tend to repeat random words, which can be verified in our case studies (Appendix E). As a result, the generated sentences are not meaningful, as shown by extremely low BLEU scores.

As shown in Figure 2c, our EqHard-EM guarantees that all decoders are trained with the same number of samples, preventing non-training collapse; since our algorithm adopts hard assignments (either 0 or 1), it also has a high variance among different samples, preventing from synchronous-training collapse. Overall, Table 2 shows that our EqHard-EM achieves the highest BLEU-F score, as well as the best diversity scores (excluding the improperly trained Hard-EM). Appendix E presents sample responses generated by Soft-EM, Hard-EM, and EqHard-EM.

Shen et al. (2019) have experimented with other EM variants including disabling the dropout and learning the prior. We analyze them in Appendix F due to the limit of space, especially as these variants are neither effective nor standard.

We further investigate whether EM-style, posterior-based assignment actually helps. We compare our EqHard-EM with two other equal-size hard assignment methods: `EqRandom-Fixed` and `EqRandom-Dynamic`. The `EqRandom-Fixed` variant pre-assigns samples to a decoder before training; this is equivalent to training each decoder on a partition of the training set. `EqRandom-Dynamic` assigns samples to decoders randomly and dynamically during training; this is essentially training a model with multiple initializations, if the number of epochs is large. In both cases, we ensure each decoder is assigned the same number of samples, so they are equal-size hard assignments. As seen, these two methods perform similarly to each other. They are slightly better than beam search, but much worse than our EqHard-EM in terms of both quality and diversity.

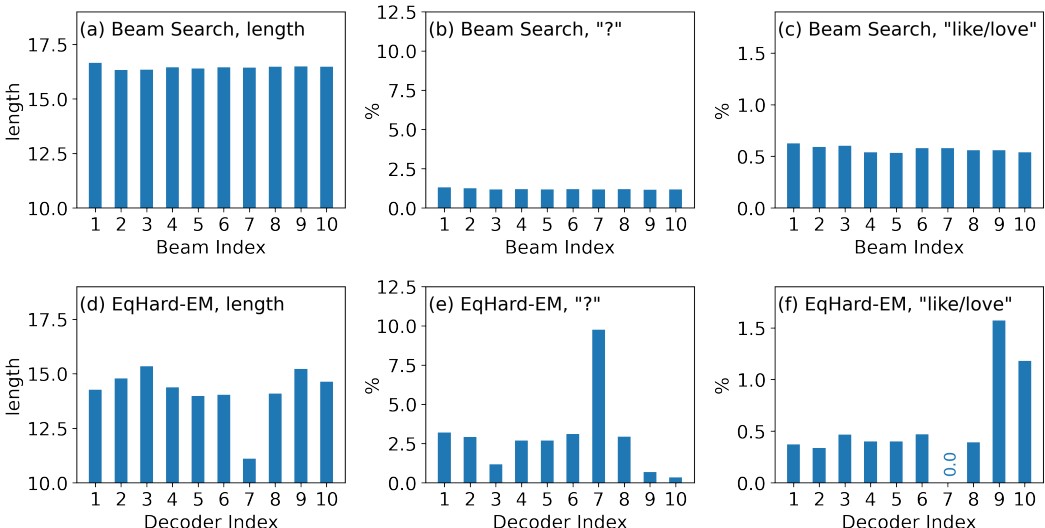

Figure 3: Comparing outputs from beam search and EqHard-EM on the Weibo dataset. Subplots (a) and (d) compare the length of generated sentences; Subplots (b) and (e) compare the percentage of the question mark; and Subplots (c) and (f) compare the percentage of the word "like/love" (喜欢) generated.

These controlled experiments convincingly show that our EqHard-EM is able to take advantage of the principled EM algorithm to learn a mixture of decoders, and that our model does not suffer from the collapse issue.

**Analyzing different decoders.** We analyze how decoders are specialized in certain aspects of dialogue responses. Admittedly, the responses generated by different decoders would not exhibit clear clustering phenomena, because our inputs are open-domain dialogue utterances covering a wide range of topics. The generated responses must fit the input in a certain way, so the between-sample variance would be larger than the decoder variance. Despite this, we do find obvious patterns of certain decoders, including the length[6], sentence structures (e.g., whether it is a question), and the preference of certain words (Figure 3). For example, Decoder 7 tends to generate short question responses, whereas Decoders 9 and 10 tend to generate positive-sentiment utterances with the word "like/love". For comparison, we also show beam search results, which are nearly uniform among different samples in the beam. Our decoder analysis, despite its simplicity, is conducted in a quantitative fashion, as opposed to previous studies that are purely based on examples (Gu et al., 2019; Shen et al., 2019). This provides rigorous evidence that our EqHard-EM is able to train multiple decoders to capture different "modes" of open-domain dialogue utterances.

**Additional results.** We present additional results in appendices. C: Effect of the number of decoders, D: Human evaluation, E: Case study, F: Analysis of additional EM variants, and G: Efficiency analysis.

## 4 RELATED WORK

Open-domain dialogue generation is a challenging task as neural dialogue systems may generate generic responses such as "I don't know", which is especially apparent for moderately sized models due to their limited modeling capacity (Li et al., 2016b; Wang et al., 2021b). In addition to the training-time and inference-time approaches mentioned in Section 1, another line of work addresses the problem using cue words to guide the response generation (Mou et al., 2016). Yao et al. (2017) enhance the gated recurrent unit to fuse cue words that are heuristically selected based on pointwise mutual information. Gao et al. (2019) jointly train a cue word inference model along with the decoder network to further improve performance. Zou et al. (2021) apply non-autoregressive

---

[6]We observe the EqHard-EM generates slightly shorter responses than beam search (Figures 3a and 3d). By examining the utterances, we find beam search tends to generate repetitive words such as "ha ha ha..."

generation to incorporate cue words in the generated response. Generic responses may also be alleviated by grounding on commonsense knowledge. Zhou et al. (2018) use a large-scale knowledge graph to ground the generation process. Wu et al. (2020) make further improvements by retrieving and fusing relevant knowledge facts. These approaches do not directly address the one-to-many phenomenon in open-domain dialogue generation, but rely on external information to make the responses more meaningful.

Large language models (LLMs) suffer less from generic responses because they have more model capacity to learn the desired distribution required for the one-to-many dialogue task. Recently, LLMs have shown success in text generation with reinforcement learning using intensive human feedback. Ouyang et al. (2022) propose InstructGPT to improve LLM's ability to follow human instructions. They warm-start the LLM by supervised learning, and then finetune it with proximal policy optimization (PPO, Schulman et al., 2017), where the feedback is provided by a reward model. Later, OpenAI releases ChatGPT, which follows a similar training pipeline and achieves remarkable performance for dialogue generation. These models require significant human annotation for both expert demonstration and reward modeling, making them inaccessible to most users and researchers. Medium-to-small-sized dialogue systems trained by supervised learning remain the common practice for dialogue generation.

Expectation–maximization (EM, Dempster et al., 1977) is a principled approach to training mixture models. The algorithm has been successfully applied in various fields (Dayan & Hinton, 1997; Sigg & Buhmann, 2008; Li et al., 2019). Specifically in natural language processing, EM is applied to unsupervised or semi-supervised training of hidden Markov models for sequential labeling (Rabiner & Juang, 1986), known as the classic Baum–Welch algorithm (Baum et al., 1970). Hofmann (1999) apply the EM algorithm for probabilistic latent semantic analysis (pLSA), where the topic is an unobserved variable. In the neural regime, the most related work to ours is Shen et al. (2019), as they apply Soft-EM and Hard-EM to diverse machine translation. We find it controversial whether diversity is needed in translation; instead, we focus on dialogue generation, where diversity is desired to avoid generic responses (Li et al., 2016a;b). Moreover, we observe neither their Soft-EM nor Hard-EM works well, and we propose a theoretically justified EqHard-EM variant.

Another related (but different) research topic is the mixture of experts (MoE, Jacobs et al., 1991; Hampshire & Waibel, 1992; Shazeer et al., 2017), where an expert $k$ predicts a real-valued feature vector $e_k$, mixed by predicted probabilities $p$ as $\sum_k p_k e_k$. The main difference is that we mix decoders' probabilities rather than features: $P(\mathbf{r}|\mathbf{c}) = \sum_k P(z = k|\mathbf{c})P(\mathbf{r}|z = k, \mathbf{c})$. However, we do not directly optimize the marginal probability because our uniform prior of $P(z|\mathbf{c})$ in Section 2 makes the problem degrade to the optimization of $P(\mathbf{r}|z = k, \mathbf{c})$ individually for different $k$s. Admittedly, we might train a model to learn the mixing probability $P(z = k|\mathbf{c})$, predicting which decoder is responsible for a context $\mathbf{c}$. This, however, violates our one-to-many assumption and indeed results in performance degradation as shown in Appendix F. By contrast, our EM formulation does not predict the decoder $z$ by merely the context $\mathbf{c}$, but determines which decoder fits a given training sample comprising both the context $\mathbf{c}$ and response $\mathbf{r}$.

## 5    CONCLUSION

In this work, we propose an Equal-size Hard Expectation–Maximization (EqHard-EM) algorithm for multi-decoder diverse dialogue generation. Our EqHard-EM adopts the hard assignment as in Hard-EM, but additionally imposes an equal-assignment constraint to ensure all decoders are well-trained. We provide detailed theoretical analysis to justify our algorithm. Extensive experiments on two large-scale datasets further confirm that our EqHard-EM algorithm enables decoders to specialize and generate high-quality and diverse responses.

**Limitation.** Our paper addresses the generic responses for dialogue systems by training a mixture of decoders. This inevitably slows down the training process similar to standard EM algorithms (Shen et al., 2019), because we need to perform inference for all decoders even though our EqHard-EM propagates gradient to one. Despite this, we find that our Hungarian algorithm has little overhead for the training process, and our approach is more efficient at inference compared with beam search, because multiple decoders can be parallelized across different GPUs.

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

## A  PROOF OF THEOREM 1

We introduce two lemmas for proving Theorem 1. The first lemma controls the error between the empirical average and the true expectation. It is a direct application of Hoeffding's inequality. The second lemma bounds the difference between the learned probability and the true underlying probability. Following the notation in Eqn. (2), we use $Q(z|\mathbf{c}, \mathbf{r})$ as the estimated posterior.

**Lemma 1.** *Given a dataset $\mathcal{D}$ consisting of context-response pairs $(\mathbf{c}, \mathbf{r})$, for any $z$, and $\epsilon > 0$, we have*

$$\Pr\left(\left|\frac{1}{|\mathcal{D}|}\sum_{(\mathbf{c},\mathbf{r})\in\mathcal{D}} Q(z|\mathbf{c}, \mathbf{r}) - \mathbb{E}_{\mathbf{c},\mathbf{r}}[Q(z|\mathbf{c}, \mathbf{r})]\right| \geq \epsilon\right) \leq 2\exp\left(-2\epsilon^2|\mathcal{D}|\right)$$

*Proof.* Observe that $Q(z|\mathbf{c}, \mathbf{r})$ is a real-valued random variable. The proof is then clear by using Hoeffding's inequality. $\qquad\square$

Lemma 1 is a quantitative description of the law of large numbers.

**Lemma 2.** *Given any $z$ and estimation $Q(z|\mathbf{c}, \mathbf{r})$ of the true $P(z|\mathbf{c}, \mathbf{r})$, we have*

$$\left| \mathbb{E}_{\mathbf{c},\mathbf{r}}[Q(z|\mathbf{c}, \mathbf{r})] - \frac{1}{K} \right| \le \mathbb{E}_{\mathbf{c},\mathbf{r}}\left[ \left| Q(z|\mathbf{c}, \mathbf{r}) - P(z|\mathbf{c}, \mathbf{r}) \right| \right]$$

*Proof.*

$$\left| \mathbb{E}_{\mathbf{c},\mathbf{r}}[Q(z|\mathbf{c}, \mathbf{r})] - \frac{1}{K} \right| = \left| \sum_{\mathbf{c}} \sum_{\mathbf{r}} Q(z|\mathbf{c}, \mathbf{r})P(\mathbf{c}, \mathbf{r}) - \frac{1}{K} \right| \tag{11}$$

$$= \left| \sum_{\mathbf{c}} \sum_{\mathbf{r}} Q(z|\mathbf{c}, \mathbf{r})P(\mathbf{c}, \mathbf{r}) - \sum_{\mathbf{c}} \sum_{\mathbf{r}} P(z|\mathbf{c}, \mathbf{r})P(\mathbf{c}, \mathbf{r}) \right| \tag{12}$$

$$= \left| \sum_{\mathbf{c}} \sum_{\mathbf{r}} P(\mathbf{c}, \mathbf{r}) \left( Q(z|\mathbf{c}, \mathbf{r}) - P(z|\mathbf{c}, \mathbf{r}) \right) \right| \tag{13}$$

$$\le \sum_{\mathbf{c}} \sum_{\mathbf{r}} P(\mathbf{c}, \mathbf{r}) \left| Q(z|\mathbf{c}, \mathbf{r}) - P(z|\mathbf{c}, \mathbf{r}) \right| \tag{14}$$

$$= \mathbb{E}_{\mathbf{c},\mathbf{r}}\left[ \left| Q(z|\mathbf{c}, \mathbf{r}) - P(z|\mathbf{c}, \mathbf{r}) \right| \right] \tag{15}$$

Here, Eqn. (12) replaces $1/K$ with the marginalization of $P(z, \mathbf{c}, \mathbf{r})$ over $\mathbf{c}$ and $\mathbf{r}$, and (14) follows the triangle inequality. $\qquad\square$

Lemma 2 bounds the absolute difference between the estimation $Q(z|\mathbf{c}, \mathbf{r})$ and the true $P(z|\mathbf{c}, \mathbf{r})$.

**Theorem 1.** *With probability at least $1 - \delta$, we have*

$$\left| \frac{1}{|\mathcal{D}|} \sum_{(\mathbf{r},\mathbf{c})\in\mathcal{D}} Q(z|\mathbf{r}, \mathbf{c}) - \frac{1}{K} \right| \le \sqrt{\frac{1}{2|\mathcal{D}|} \log\left(\frac{2K}{\delta}\right)} + \mathbb{E}_{\mathbf{c},\mathbf{r}}\left[ \left| Q(z|\mathbf{c}, \mathbf{r}) - p(z|\mathbf{c}, \mathbf{r}) \right| \right]$$

*for all $z \in \{1, \cdots, K\}$. Here, $Q(z|\mathbf{c}, \mathbf{r})$ is an estimated posterior and $\mathcal{D}$ is a dataset with finite samples.*

*Proof.* Lemma 1 shows an inequality that upper bounds the deviation from a uniform assignment for a particular $z$. The probability for the original inequality to hold for all $z$ is then:

$$\Pr\left( \forall z \in [K] : \left| \frac{1}{|\mathcal{D}|} \sum_{(\mathbf{c},\mathbf{r})\in\mathcal{D}} Q(z|\mathbf{c}, \mathbf{r}) - \mathbb{E}_{\mathbf{c},\mathbf{r}}[Q(z|\mathbf{c}, \mathbf{r})] \right| \ge \epsilon \right) \le 2K \exp\left( -2\epsilon^2 |\mathcal{D}| \right)$$

Choose an $\epsilon$ such that $\delta = 2K \exp\left( -2\epsilon^2 |\mathcal{D}| \right)$. We have

$$\Pr\left( \forall z \in [K] : \left| \frac{1}{|\mathcal{D}|} \sum_{(\mathbf{c},\mathbf{r})\in\mathcal{D}} Q(z|\mathbf{c}, \mathbf{r}) - \mathbb{E}_{\mathbf{c},\mathbf{r}}[Q(z|\mathbf{c}, \mathbf{r})] \right| \ge \sqrt{\frac{1}{2|\mathcal{D}|} \log\left(\frac{2K}{\delta}\right)} \right) \le \delta$$

Then with probability at least $1 - \delta$, we have:

$$\left| \frac{1}{|\mathcal{D}|} \sum_{(\mathbf{c},\mathbf{r}) \in \mathcal{D}} Q(z|\mathbf{c},\mathbf{r}) - \frac{1}{K} \right|$$

$$\leq \left| \frac{1}{|\mathcal{D}|} \sum_{(\mathbf{r},\mathbf{c}) \in \mathcal{D}} Q(z|\mathbf{c},\mathbf{r}) - \mathbb{E}_{\mathbf{c},\mathbf{r}}[Q(z|\mathbf{c},\mathbf{r})] \right| + \left| \mathbb{E}_{\mathbf{c},\mathbf{r}}[Q(z|\mathbf{c},\mathbf{r})] - \frac{1}{K} \right|$$

$$\leq \sqrt{\frac{1}{2|\mathcal{D}|} \log\left(\frac{2K}{\delta}\right)} + \left| \mathbb{E}_{\mathbf{c},\mathbf{r}}[Q(z|\mathbf{c},\mathbf{r})] - \frac{1}{K} \right| \qquad \text{(w.p. } 1 - \delta \text{, by Lemma 1)}$$

$$\leq \sqrt{\frac{1}{2|\mathcal{D}|} \log\left(\frac{2K}{\delta}\right)} + \mathbb{E}_{\mathbf{c},\mathbf{r}}\left[ \left| Q(z|\mathbf{c},\mathbf{r}) - P(z|\mathbf{c},\mathbf{r}) \right| \right] \qquad \text{(by Lemma 2)}$$

concluding the proof. $\qquad\qquad\square$

## B    IMPLEMENTATION DETAILS

Our multi-adapter model used T5-small (Raffel et al., 2020) from HuggingFace (Wolf et al., 2019) as the backbone: 6 layers, 8 attention heads, an input/output dimension of 512, and an inner dimension of 2048 for both the encoder and decoder. We implemented our multi-decoder model by inserting an adapter layer with an inner dimension of 256 after each Transformer decoder layer. We adopted 10 decoders based on the common evaluation metric of diverse text generation (Zhao et al., 2017; Gu et al., 2019). Table 3 shows a comparison between our multi-adapter architecture and the full-Transformer architecture. As seen, our multi-adapter model retains a competitive BLEU performance while only using 24% of the parameters. We adopted the multi-adapter architecture in the main experiments, because it helped our development process much.

For training, we had two stages: 1) Pretraining of the shared Transformer model, and 2) EqHard-EM training of the adapter layers. In both stages, we used a learning rate of 0.001 with the Adam optimizer with $\boldsymbol{\beta} = (0.9, 0.999)$. We used a batch size of 64 for pretraining. For our 10-decoder EqHard-EM training, we considered 640 samples in an E-step, assigning 64 samples for each decoder.

## C    EFFECT OF THE NUMBER OF DECODERS

We investigate how well our approach scales with the number of decoders. Figure 4 shows a comparison between beam search and our multi-decoder model under 5-, 10-, and 15-generation settings. Due to the limited resources, we conducted the analysis only on the Weibo dataset. As seen, our multi-decoder model performs similarly to beam search when there are only 5 decoders. However, the performance gap increases when we have more decoders, which verifies the need for our multi-decoder model.

## D    HUMAN EVALUATION

We conducted human evaluation to compare both the quality and diversity of the responses from T5-small (beam search) and our multi-decoder model. We again chose the Weibo dataset, because it is less noisy. Five native Chinese speakers were provided with a dialogue context and generated responses of two systems, and were asked to select the better system or indicate a tie. For quality

| Decoder Architecture | #Parameters | BLEU2-F |
|---|---|---|
| Adapter | 108M | 10.13 |
| Full Transformer | 451M | 11.08 |

Table 3: Comparison between adapter and full-Transformer architectures for our 10-decoder model.

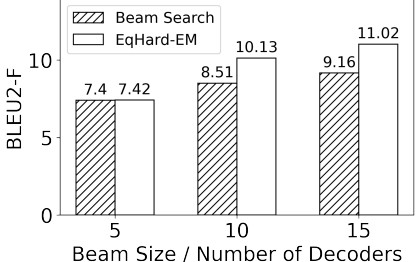

Figure 4: Comparison between beam search and our multi-decoder model in 5-, 10-, and 15-generation settings on the Weibo dataset.

| | Model | Wins | Ties | Losses | $p$-value |
|---|---|---|---|---|---|
| Overall Quality | T5-small beam search | 32.8% | 19.0% | 48.2% | <0.0001 |
| | Multi-adapter + EqHard-EM | **48.2%** | 19.0% | **32.8%** | |
| Diversity | T5-small beam search | 4.8% | 8.0% | 87.2% | <0.0001 |
| | Multi-adapter + EqHard-EM | **87.2%** | 8.0% | **4.8%** | |

Table 4: Human evaluation for the overall quality and diversity of the generated responses. Five annotators were invited to evaluate 100 samples on the Weibo dataset. The $p$-value is given by the one-sided binomial test (ignoring ties).

evaluation, annotators were presented with only one response from each model, preventing them from identifying the systems based on diversity. For diversity, all 10 responses were presented. In both cases, the presentation order of the two systems was randomly shuffled, so the annotators could not tell which system generated which response.

Table 4 shows the results of human evaluation. Our multi-decoder model shows statistically significant improvement in terms of overall quality and diversity. Especially, annotators believe our approach generates more diverse utterances on 18x more samples. The results are highly consistent with the automatic metrics in Table 4, confirming the effectiveness of our approach.

## E  CASE STUDY

Table 5 shows example responses from Weibo and OpenSubtitles. As seen, Soft-EM exhibits the synchronous-training collapse, where all decoders generate highly similar responses. Hard-EM, on the other hand, suffers from the non-training collapse, where a large number of decoders are not properly trained and generate low-quality text (e.g., certain words being repeated). By contrast, our EqHard-EM algorithm enables us to properly train all decoders to generate high-quality, diverse responses.

## F  ANALYSIS OF ADDITIONAL EM VARIANTS

Shen et al. (2019) also realize the collapse issues of classic EM algorithms; however, they explain the poor performance of Soft-EM by drawing an analogy to the bypassing of latent variables in variational auto-encoders (VAE, Bowman et al., 2016). Based on such interpretation, they propose a no-dropout trick that disables the dropout layers for the E-step while retaining them for the M-step.

We analyze the no-dropout trick in Table 6. As seen, disabling dropout has minimum effect on the EM algorithms, and does not prevent collapses for Hard-EM and Soft-EM. This is not surprising because even Shen et al. (2019) acknowledge that their no-dropout trick does not work consistently, as shown in Table 7 of Appendix C in their paper (red and blue numbers). The one-to-many phenomenon in our dialogue setting is more severe than their machine translation setting, and it is understandable that such an ad hoc trick, without much theoretical justification, may fail in the more difficult dialogue task.

We also analyze EM variants with learned priors, where a neural network is used to predict the decoder given the dialogue context. This is another variant in Shen et al. (2019). As shown, learned

| Weibo Input: 终于买了个爱疯，发觉也没啥用，还不如ipad | | |
|---|---|---|
| *Finally got an iPhone, found it useless, even worse than an ipad* | | |
| Soft-EM | Hard-EM | EqHard-EM |
| 我也有一个，不过没有那么大 | 江江江[repeat] | 这个是什么牌子的？ |
| *I have one too, but not so big* | *river river river [repeat]* | *What brand is this?* |
| 我也有一个，不过没有那么大 | [eos] | 我也想买一个，可惜没钱买 |
| *I have one too, but not so big* | *[eos]* | *I want to buy one too, too bad I don't have money* |
| 我也有一个，不过没有那么大 | 珍珍珍[repeat] | 我也是，还不如ipad呢 |
| *I have one too, but not so big* | *treasure treasure treasure [repeat]* | *Me too, even worse than an iPad indeed* |
| 我也有一个，不过没有那么大 | [eos] | 我也想买一个，可惜没钱买 |
| *I have one too, but not so big* | *[eos]* | *I want to buy one too, too bad I don't have money* |
| 我也有一个，不过没有那么大 | [eos] | 这个东意不错，我也想买一个 |
| *I have one too, but not so big* | *[eos]* | *This stuff is good, I want to buy one too* |
| 我也有一个，不过没有那么大 | 我也有一个，不过没有那么大 | 我也有一个，不过没有ipad |
| *I have one too, but not so big* | *I have one too, but not so big* | *I have one too, but not the ipad* |
| 我也有一个，不过没有那么大 | 想想想[repeat] | 你这是在哪里啊？我也想买一个 |
| *I have one too, but not so big* | *think think think [repeat]* | *Where are you at? I want to buy one too* |
| 这个是什么？求解 | 苇苇苇[repeat] | 我也想买一个，但是没钱买 |
| *What is this? Please exp[lain]* | *reed reed reed [repeat]* | *I want to buy one too, but I don't have money* |
| 我也有一个，不过没有那么大 | 雪雪雪[repeat] | 我也买了一个，不过还是很不错的 |
| *I have one too, but not so big* | *snow snow snow [repeat]* | *I bought one too, but it's actually pretty good* |
| 这个是什么？求解释 | 。。。[repeat] | 这个不错，不过我喜欢。 |
| *What is this? Please explain* | *... [repeat]* | *This is not bad, but I like it.* |
| OpenSubtitles Input: and you know what happens if we object? | | |
| Soft-EM | Hard-EM | EqHard-EM |
| i'm not gonna object. | what what what [repeat] | i don't know. |
| i do. | i'm not a victim. | what do you mean? |
| i don't know. | and and and [repeat] | if we object, we 'll be able to. |
| i don't know. | if really really really [repeat] | i don't know, i don't know. |
| i do. | no pe pe pe [repeat] | i don't know. |
| i don't know. | we're gonna have to do this. | i'm not gonna be able to do anything. |
| i don't know. | okay okay okay [repeat] | we're gonna have to. |
| i don't know. | hurry hurry hurry [repeat] | what? |
| i do. | what? | i don't know. |
| i don't know. | then then then [repeat] | i'm not gonna be able to do this, mr. santos. |

Table 5: Case studies. Translations are in *italic*. [repeat] indicates that further repetitions are omitted, and [eos] indicates an end-of-sentence token is generated as the first token.

priors perform worse than uniform priors due to the more severe collapse problem, which is demonstrated by the extremely low BLEU scores. This is because a learned prior gives the model more opportunities to select the best decoder, further worsening the "rich-gets-richer" effect.

Therefore, we do not consider the recurrent dropout trick and learned priors in our main experiments due to their ineffectiveness and poor performance.

## G  EFFICIENCY ANALYSIS

We evaluated training and inference efficiency on an i9-9900X CPU and a TITAN RTX GPU. Compared with classic EM variants, our EqHard-EM involves an additional assignment step using the Hungarian algorithm. As shown in Table 7: the E-step, Hungarian algorithm, and M-step take 75.69%, 0.97%, and 24.31% of the total training time, respectively. This suggests that our EqHard-EM incurs a negligible overhead compared with soft and hard EM variants. At inference, our multi-adapter model processes 79.1 samples per second, whereas beam search processes 16.9 samples per second. Our approach achieves a 4.7x speedup because inference for each decoder can be computed in parallel across multiple GPUs, whereas beam search cannot leverage the same level of parallelism.

| Dataset | Model | | BLEU1 | | | BLEU2 | | | Distinct n-gram | | Pairwise |
|---|---|---|---|---|---|---|---|---|---|---|---|
| | | | F | P | R | F | P | R | Dist-1 | Dist-2 | BLEU |
| Weibo | Soft-EM | uniform prior | 17.02 | **27.55** | 12.32 | 6.27 | **16.03** | 3.89 | 10.69 | 12.45 | 86.25 |
| | | uniform prior, no dropout | 17.13 | 27.53 | 12.43 | 6.30 | 16.02 | 3.92 | 10.88 | 12.77 | 84.77 |
| | | learned prior | 10.33 | 8.63 | 12.86 | 3.18 | 2.76 | 3.74 | 27.89 | 54.53 | 1.09 |
| | | learned prior, no dropout | 13.58 | 12.95 | 14.27 | 4.50 | 4.66 | 4.35 | 20.56 | 38.31 | 4.02 |
| | Hard-EM | uniform prior | 4.81 | 3.09 | 10.84 | 2.15 | 1.61 | 3.22 | 4.01 | 4.11 | 0.00 |
| | | uniform prior, no dropout | 5.17 | 3.39 | 10.86 | 2.25 | 1.71 | 3.27 | 3.51 | 3.60 | 2.22 |
| | | learned prior | 5.07 | 3.30 | 10.92 | 2.14 | 1.61 | 3.20 | 5.22 | 5.38 | 0.00 |
| | | learned prior, no dropout | 4.83 | 3.11 | 10.81 | 2.32 | 1.78 | 3.31 | 9.65 | 10.02 | 0.02 |
| | | EqHard-EM | **22.65** | 26.17 | **19.96** | **10.13** | 14.55 | **7.77** | 27.03 | 42.02 | 22.32 |
| Open-Subtitles | Soft-EM | uniform prior | 15.48 | **14.55** | 16.54 | 3.03 | **2.76** | 3.35 | 12.89 | 15.35 | 72.95 |
| | | uniform prior, no dropout | 15.52 | 14.51 | 16.67 | 3.00 | 2.71 | 3.37 | 13.23 | 15.93 | 70.29 |
| | | learned prior | 15.45 | 14.52 | 16.50 | 3.01 | **2.76** | 3.31 | 13.01 | 15.45 | 71.86 |
| | | learned prior, no dropout | 15.47 | 14.48 | 16.60 | 2.95 | 2.71 | 3.25 | 13.40 | 16.17 | 67.46 |
| | Hard-EM | uniform prior | 6.84 | 4.13 | 19.81 | 1.25 | 0.73 | 4.30 | 4.83 | 4.81 | 1.50 |
| | | uniform prior, no dropout | 9.20 | 5.66 | 24.54 | 1.56 | 0.88 | 6.82 | 9.87 | 10.60 | 0.61 |
| | | learned prior | 4.71 | 2.73 | 17.11 | 0.81 | 0.46 | 3.31 | 6.82 | 5.57 | 0.23 |
| | | learned prior, no dropout | 4.70 | 2.67 | 19.85 | 0.70 | 0.38 | 3.80 | 8.60 | 6.57 | 0.00 |
| | | EqHard-EM | **17.86** | 12.98 | **28.60** | **3.76** | 2.36 | **9.27** | 34.09 | 50.41 | 13.15 |

Table 6: Comparison with additional EM variants on Weibo and OpenSubtitles.

| Training Time | | |
|---|---|---|
| E-Step | M-Step | Hungarian |
| 75.69% | 24.31% | 0.97% |
| Inference Time | | |
| Beam search | 16.9 samples per second | |
| Ours | 79.1 samples per second | |

Table 7: Training and inference efficiency analysis. The top half shows the percentage of training time spent in the E-step, M-step, and the Hungarian algorithm (part of the E-step). The bottom half compares the inference efficiency of our EqHard-EM algorithm against beam search.

ACKNOWLEDGMENTS

We would like to thank all reviewers and chairs for their comments. We would also like to thank Nidhi Hegde and Herbert Yang for their valuable suggestions. This research was supported in part by Natural Sciences and Engineering Research Council of Canada (NSERC) under Grant No. RGPIN2020-04465, the Amii Fellow Program, the Canada CIFAR AI Chair Program, the Digital Research Alliance of Canada (alliancecan.ca), and a Mitacs Accelerate (Cluster) grant.

