# OpenReview forum: "An Equal-Size Hard EM Algorithm for Diverse Dialogue Generation"
_ICLR.cc/2023/Conference — ICLR 2023 poster_

### Official Review · Reviewer_3nYS · 2022-10-24

**Confidence:** 3
**Correctness:** 3
**Technical Novelty And Significance:** 2
**Empirical Novelty And Significance:** 2
**Recommendation:** 3

**Clarity, Quality, Novelty And Reproducibility:**

The authors should publicize their code, and testing results, which are not stated in the paper.

**Strength And Weaknesses:**

Strengths: The idea is clear and the implementation of the idea should be easy.

Weakness:
The authors argue that the E-step in the proposed model contributes to less than 1% of the total training time. I want to know if the full experimental results can be provided, including the total training time, the number of EM steps, and the exact time of the E-step and M-step in the training time. Also, the time efficiency should be compared for all methods in your experiments.

The experimental results are not convincing.  First, only results on BLEU-2 of BLEU-n are given. How about BLEU-1? Also, human evaluation results of Table1 should be given.  Also, I think a pretrained T5 should not be proper to compare here, as it is not for dialog generation. Perhaps DialogGPT is better if the authors want to compare a pre-trained model. However, I found most of the compared methods except the proposed one perform even worse than T5, which is hard to believe.

**Summary Of The Paper:**

This paper addresses the issue of generic response generation by first using adapters for efficient multi-decoder frameworks, and next using a balanced EM algorithm to train the EM of the multi-decoders. The main contribution is to train the balanced EM by the classic Hungarian algorithm. Experiments compare the proposed model with other methods that address the same problem.


**Summary Of The Review:**

Overall, I think the idea is reasonable but the experiments are a bit weak.

---

> ### Author Response · Authors · 2022-11-09
> **Response to reviewer 3nYS (Part 1)**
>
> The reviewer believes our idea is reasonable, but mainly complains about our experiments. We find that they are largely due to misunderstandings.
>
> > Weakness: "The authors argue that the E-step in the proposed model contributes to less than 1% of the total training time. I want to know if the full experimental results can be provided, including the total training time, the number of EM steps, and the exact time of the E-step and M-step in the training time. Also, the time efficiency should be compared for all methods in your experiments."
>
> This is a misunderstanding about our claim. As mentioned in the Approach section, we stated that the Hungarian algorithm takes 1% of the total time, which does not have much overhead. The E-step requires a forward pass, which takes more time.
>
> We thank the reviewer for the suggestion, and have evaluated the time efficiency in a quantitative manner on an i9-9900X CPU and a TITAN RTX GPU:
>
> Training Time: E-step: 75.69%, the Hungarian algorithm within the E-step: 0.97%, and the M-step: 24.31%.
>
> Inference time: Our EM approach:  79.1 samples per second. Beam search of T5: 16.9 samples per second. Our approach, in fact, achieves a 4.7x speedup because inference for each decoder can be computed in parallel across multiple GPUs, whereas beam search cannot leverage the same level of parallelism.
>
> We did not report detailed efficiency results because our main contribution is to propose EqHard-EM to address the training collapse issues, but all EM variants have the same efficiency. Moreover, the tables in our main paper have already been too wide.
>
> In the revision, we nevertheless mention training/inference speed in the newly added reproducibility section to address the reviewer's concern.
>
> > Weakness: "only results on BLEU-2 of BLEU-n are given. How about BLEU-1?"
>
> Thanks for the suggestion. In the original version, we followed [Liu et al.](https://aclanthology.org/D16-1230.pdf) and used BLEU-2. We did not report BLEU-1 due to space limit. In our revision, we used a smaller font to include BLEU-1, as per your suggestion. Results show that our approach is consistently better than other baseline methods in BLEU-1 as well.
>
> > Weakness: "I think a pretrained T5 should not be proper to compare here, as it is not for dialog generation. Perhaps DialogGPT is better if the authors want to compare a pre-trained model".
>
>
> The pretrained T5 model is finetuned (second-stage pretrained) on the large dialogue datasets before we apply EqHard-EM, clearly stated at the beginning of Sec 3.2 and Appendix B. T5 is known for its ability to generalize, as demonstrated by its superb performance in a large number of text generation tasks. We further finetune (second-stage pretrain) it on dialogue datasets, so that our T5 is fully aware of the dialogue generation task.
>
> In our experiments, we used the same finetuned T5 as the backbone for all our model variants, establishing a fair comparison between strong baseline models and our approach.
>
> DialoGPT is problematic in our case because it is a decoder-only model, which does not explicitly model the **one**-to-many process. In our preliminary study, we tried GPT-2, but it achieved lower performance than T5 (Footnote 3 in our revision). We also compared our approach against a recently proposed DialogBERT model (Table 1), and show that EqHard-EM achieves significant improvement over it.

---

> > ### Author Response · Authors · 2022-11-09
> > **Response to reviewer 3nYS (Part 2)**
> >
> > > Weakness: "I found most of the compared methods except the proposed one perform even worse than T5, which is hard to believe."
> >
> > All our experiments have been rigorously conducted and we have presented all results and conclusions in an earnest way.
> >
> > T5-small being the second best model is, in fact, understandable because
> >
> > 1) Alleged state-of-the-art dialogue models do not outperform finetuning a pretrained language model on deduplicated dialogue datasets. A very [recent study](https://aclanthology.org/2022.lrec-1.16.pdf) [1] shows that a number of existing dialogue datasets have overlapping samples (up to 30%) among training, validation, and test splits. After deduplicating, those alleged state-of-the-art models fail to achieve comparable performance in the setting of one-response generation, shown by the above study. Our work focuses on diverse generation, and we achieved consistent results in the setting of multi-response generation: alleged state-of-the-art models are simply worse than finetuning a pretrained language model on deduplicated dialogue datasets.
> >
> > 2) Traditional EM algorithms do not outperform a plain Seq2Seq model in terms of text quality. This was realized by Shen et al. (2020) in Table 2 of their paper, but they did not mention it in Abstract, Intro, or Conclusion sections. In our work, we implemented Soft-EM and Hard-EM in the same framework as our EqHard-EM. Our results are consistent to Shen et al. (2020), showing that traditional EM algorithms are worse than a plain Seq2Seq model.
> >
> > We share the same concern with the reviewer that a number of previous studies are wrong (e.g., using significantly overlapping data). Therefore, we have conducted our experiments rigorously and published our code (anonymously) even during the double-blind review phase. We followed the correct setting for scientific research, and have also clearly and consistently stated the findings throughout our paper.
> >
> > [1] Wen, Y., Luo, G. and Mou, L., 2022. An Empirical Study on the Overlapping Problem of Open-Domain Dialogue Datasets. LREC2022.
> >
> > > Clarity, Quality, Novelty And Reproducibility: "The authors should publicize their code, and testing results, which are not stated in the paper."
> >
> > As stated in Footnote 1, Page 1, we released our code at https://github.com/anonymous-1759/EqHard-EM.
> >
> > We understand that this link might be confusing, because it is anonymized for double-blind review. However, this is a real repo.
> >
> > ---
> >
> > Summary of our response:
> >
> > The reviewer gave an extraordinary score, even though the reviewer recognized our idea as being "reasonable".
> >
> > In fact, the reviewer had several misunderstandings (missing footnote 1 for the code repo, the confusion of time calculation), and we have clarified them in the response and the revision.
> >
> > The reviewer also had concerns about the efficiency, but our approach turns out to be more efficient than a plain seq2seq model.
> >
> > It is understandable that the reviewer might be misled by previous papers (which use overlapping datasets). In our author response, we have clearly pointed to the evidence in previous work showing the pitfall. We sincerely hope that the reviewer can read our response, check previous papers, and render a more convincing judgment about our work.

---

> > > ### Author Response · Authors · 2022-11-16
> > > **Followup response to reviewer 3nYS**
> > >
> > > The reviewer believes that the E-step of our proposed model contributes to less than 1% of the total training time. However, this is a misunderstanding. We have clearly stated that it’s the Hungarian algorithm that contributes to less than 1% of the total training time. Due to the reviewer’s confusion, we have conducted additional experiments and provided detailed training and inference time efficiencies of our algorithm. In addition, the reviewer believes that we have not released our source code, which is false. In fact, we have been actively updating our anonymous GitHub repository as we conduct more experiments suggested by the reviewers.
> > >
> > > According to the ICLR guideline, the author-reviewer discussion is due in a few days. We urge the reviewer to read our response and re-evaluate our work. Thank you!

---

### Official Review · Reviewer_ov7h · 2022-10-24

**Confidence:** 4
**Correctness:** 2
**Technical Novelty And Significance:** 2
**Empirical Novelty And Significance:** 2
**Recommendation:** 5

**Clarity, Quality, Novelty And Reproducibility:**

Please refer to Detailed Comments.


**Strength And Weaknesses:**

Strengths:
- This paper explores mixture models for dialogue generation tasks.
- This paper proposes EqHard-EM to handle the non-training collapse of the Hard-EM algorithm.

Weaknesses:
- It is an incremental work of Shen et al., (2019) and the novelty of the proposed method is limited.
- It is fine to apply and extend the method of Shen et al., (2019) in dialogue generation tasks, but the experiments are not good enough, including implementing all model variants in Shen et al., (2019) and full human evaluation.
- The main contribution of this paper is to introduce equal-assignment constraints to handle the non-training collapse of the Hard-EM algorithm. However, the E-step of Hard-EM in this paper is different from Shen et al., (2019), which leverages learned prior (lp) or uniform prior (up) to ease this issue. It is difficult to judge that this method could lead to further improvements compared with Shen et al., (2019)'s version of Hard-EM.


Detailed Comments:

Overall, this paper is easy to follow and this idea is straightforward. I like this idea of introducing mixture models for diverse dialogue generation. But the experiments are not convincing enough to me. The E-step of Hard-EM in this paper is different from Shen et al., (2019), which leverages learned prior (lp) or uniform prior (up) to ease the non-training collapse of the Hard-EM algorithm. It is not sure that this method could lead to further improvements when combined with the version of the Hard-EM algorithm of Shen et al., (2019). It is necessary to reproduce the version of Shen et al., (2019) in the dialogue generation task.

While Shen et al., (2019) have explored four model variants and some crucial tricks (e.g., the effect of dropout) for diverse machine translation, it is better to revisit these models and details in dialogue generation. In addition, this paper lacks full human evaluations across two datasets and different methods, and automatic evaluations are not convincing enough in dialogue generation tasks. The authors only perform human evaluations on beam search and the proposed method, which should be improved.


Questions for the Author(s):
- As shown in Table 5, the example of the Hard-EM algorithm tends to repeat. Since we directly optimize the conditional language model in M-step, these examples are weird. What is your thought about it?
- Why is the E-step of the Hard-EM algorithm in this paper different from Shen et al., (2019)? ( i.e.,  argmax_{z} p (z| x,y; \theta) v.s.  argmax_{z} p (y| z,y; \theta)  )
- Have you evaluated the effect of the dropout during the EM algorithm?
- How about training different decoders instead of introducing adapter layers?



**Summary Of The Paper:**

This paper explores mixture models for diverse dialogue generation, following the same pipeline proposed by Shen et al., (2019). This method adopts a multi-decoder model to address the one-to-many mapping phenomenon, where a multi-adapter architecture is used to form the multi-decoder model. Based on that, the EM algorithm (soft or hard) is leveraged to learn the latent variable (i.e.,  the choice of the decoder). The authors further handle the non-training collapse of the Hard-EM algorithm by introducing equal-assignment constraints, which ensure that all decoders are adequately trained. Automatic evaluations on Weibo and OpenSubtitles datasets prove the effectiveness of introduced equal-assignment constraints, compared with the original EM algorithm.

**Summary Of The Review:**

This paper investigates introducing mixture models for diverse dialogue generation and proposes EqHard-EM to handle the non-training collapse of the Hard-EM algorithm. But the experiments are not convincing enough to me, including lacking more details of all model variants in Shen et al., (2019) and full human evaluation. The E-step of Hard-EM in this paper is also different from Shen et al., (2019), and it is uncertain whether this method will be further improved when combined with the version of Shen et al., (2019).

---

> ### Author Response · Authors · 2022-11-08
> **Response to reviewer ov7h (Part 1)**
>
> We thank the reviewer for "like[ing] this idea of introducing mixture models for diverse dialogue generation".
>
> > Weakness (1). "It is an incremental work of Shen et al., (2019) and the novelty of the proposed method is limited."
>
> By saying “the EM algorithm (soft or hard) is leveraged to learn the latent variable”, the reviewer believes our work is incremental because our main contribution is to follow Shen et al. for EM text generation.
>
> However, this is not true, as our claimed contribution is the EqHard-EM algorithm only. In our paper, soft-EM and hard-EM are presented in order to motivate our EqHard-EM approach, and are **not** part of our approach (and in fact soft-EM and hard-EM barely work in our application).
>
> The contribution of the proposed EqHardEM itself is significant because:
>
> (1)  We proposed a novel and theoretically understood EM algorithm, whereas Shen et al. published an ICML paper by applying existing EM algorithms (soft-EM and hard-EM) to text generation, and performed an experimental study with ad hoc tricks (e.g., removing dropout, which actually does not work well in our application). Our work is well motivated by a thorough literature study and empirical evidence, and we developed two lemmas and a theorem to justify our equal-assignment constraint. The novelty is recognized by all other reviewers.
>
> (2) Compared with traditional machine learning studies (e.g., a classic [EM-variant paper](https://www.cs.toronto.edu/~radford/ftp/emk.pdf) [1]) only validating the approach on mixture-of-Gaussian data), we validated our novel EM algorithm in a real-world application with several large-scale datasets.
>
> Therefore, our work is substantial in both technical novelty and empirical efforts, in comparison with previous literature.
>
> [1] Neal, R.M. and Hinton, G.E., 1998. A view of the EM algorithm that justifies incremental, sparse, and other variants. Learning in Graphical Models.
>
> > Weakness (2). "the experiments are not good enough, including implementing all model variants in Shen et al., (2019) and full human evaluation."
>
>
> (all model variants in Shen et al. (2019))
>
> Shen et al.'s paper is an empirical study. They tried various configurations and concluded that their variants are “offering slightly different trade-offs between quality and diversity” (the second last paragraph before their Footnote 6). Since we have already overflowed to the appendix, we will not be able to include all their variants as baselines, or otherwise, the volume of this paper will be more than doubled.
>
> We have nevertheless included standard soft-EM and hard-EM, which are also suggested by Shen et al. as strong variants. In addition, we tried Shen et al.’s own variant (removing dropout) in our pilot experiments and observed no improvement. Even Shen et al. themselves realized the ineffectiveness of their approaches: all their EM variants achieve a lower BLEU than a plain seq2seq model (Table 2 in their paper). By contrast, our EqHard-EM improves both quality and diversity, shown by both automatic metrics and human evaluations.
>
> We’re currently conducting the experiment of removing dropout again and will update when we obtain the results.
>
>
> (full human evaluation)
>
> The reviewer suggests conducting a full human evaluation on all model variants. This, unfortunately, is infeasible again, because each human annotator would have to read 18,000 utterances: 2 datasets x 9 variants x 100 samples x 10 utterances/sample. This not only puts a huge burden on human annotators, but also degrades the human annotation quality (as no humans can patiently read ~20K utterances).
>
> Instead, we selected a strong competing method and conducted pairwise human evaluation, which shows that our model outperforms the competing method by a large margin (48.2% vs 32.8% in terms of overall quality and 87.2% vs 4.8% in terms of diversity). The results are highly consistent with automatic metrics as well: modest improvement in BLEU scores and very large improvements in diversity scores.
>
> In summary, our experimental design has due consideration of the efficacy and feasibility. Both automatic metrics and human evaluation consistently and convincingly show the effectiveness of our EqHard-EM algorithm.

---

> > ### Author Response · Authors · 2022-11-08
> > **Response to reviewer ov7h (Part 2)**
> >
> > > Weakness (3). "The main contribution of this paper is to introduce equal-assignment constraints to handle the non-training collapse of the Hard-EM algorithm. However, the E-step of Hard-EM in this paper is different from Shen et al., (2019), which leverages learned prior (lp) or uniform prior (up) to ease this issue. It is difficult to judge that this method could lead to further improvements compared with Shen et al., (2019)'s version of Hard-EM."
> >
> > The reviewer believes our E-step is different from Shen et al., which we kindly point out is a misunderstanding. Our E-step is the same as Shen et al.’s version with the uniform-prior assumption.
> >
> > In a standard EM (i.e., soft EM), the assignment is given by the posterior $p(z|y, x)\propto p(z,y|x)=p(z|x)p(y|z,x)$. In the hard EM, this becomes $z_*=\operatorname{argmax}_z p(z|y,x)=\operatorname{argmax}_z p(z,y|x)$.
> >
> > In our work, we have the uniform prior assumption given any x, so $p(z|y,x)$ is further proportional to $p(y|z,x)$. It is noted that our notations are different: $x$ is $c$ (context) and $y$ is $r$ (response). Therefore, we actually have the same form, as shown by Eqn. (4). In the revision, we included further clarification under Eqn. (4).
> >
> > Shen et al. additionally tried a learned prior, which is not different from the uniform prior because:
> >
> > 1) Shen et al. report that all models work similarly, “offering slightly different trade-offs between quality and diversity” (and all their variants are worse than a plain seq2seq model in BLEU scores, shown in their Table 2), but our EqHard-EM yields improvement in both BLEU and diversity metrics.
> >
> > 2) Our assignment statistics (Fig 2) suggest that the learned prior will just be uniform for soft-EM, but one-hot for hard-EM. Therefore, not much is different.
> >
> > Please also note that the learned prior does not fit our EqHard-EM, which relies on the uniform-prior assumption. We will try to implement the learned prior after the dropout experiment (see Weakness 2).
> >
> > > Question (1): "As shown in Table 5, the example of the Hard-EM algorithm tends to repeat. Since we directly optimize the conditional language model in M-step, these examples are weird. What is your thought about it?"
> >
> > This is not weird but is highly expected, because our decoders are inserted with randomly initialized adapter layers. In the Hard-EM variant, we observe severe non-training collapse (shown by Figure 2), where most decoders are not properly trained and their outputs are of low quality. We have explained this in Appendix D.
> >
> > > Question (2) "Why is the E-step of the Hard-EM algorithm in this paper different from Shen et al., (2019)? ( i.e., argmax_{z} p (z| x,y; \theta) v.s. argmax_{z} p (y| z,y; \theta) )"
> >
> > As mentioned in the response to Weakness (3), they are equivalent, which can be shown by Bayes' rule. In short, the posterior is proportional to the likelihood, and hence they are the same after taking the argmax. We have clarified this in the revision after Eqn (4).
> >
> > > Question (3): "Have you evaluated the effect of the dropout during the EM algorithm?"
> >
> > Yes. At the development stage of this research, we tried the "recurrent dropout trick". It did not prevent collapse in our setting, so we did not follow their trick. We are currently running experiments for this setting again, and will give an update when the results are ready.
> >
> > > Question (4): "How about training different decoders instead of introducing adapter layers?"
> >
> > This experiment has already been shown in Table 3 in the Appendix, also mentioned in Paragraph 2, Section 3.1. It shows that using full-Transformer decoders yields a slight improvement in performance at the cost of several times more parameters.
> >
> > Our main experiments are conducted on the multi-adapter architecture because we have a large number of competing variants in the ablation study and we need to save multiple checkpoints per run.
> >
> > ---
> >
> > Summary of our response:
> >
> > Thanks for the detailed comments. The reviewer appears to have some misunderstanding about the E-step. We have clarified in the response and our revision.
> >
> > It would be much appreciated if the reviewer could double-check the details as well as our significance (both technical depth and experimental efforts) in a fair comparison with previously published work.

---

> > > ### Author Response · Authors · 2022-11-11
> > > **Followup reponse to reviewer ov7h**
> > >
> > > Thanks again for your detailed comments. We have conducted additional experiments to evaluate the effect of recurrent dropout, and we have obtained the following results on the Weibo dataset. The results are consistent with our pilot study.
> > >
> > > | Model | BLEU-1F | BLEU-2F | Dist-1 | Dist-2 | Pairwise-BLEU |
> > > |---|---|---|---|---|---|
> > > | Hard-EM | 4.81 | 2.15 | 4.01 | 4.11 | 0.00 |
> > > | Hard-EM w/ Dropout Trick | 5.17 | 2.25 | 3.51 | 3.60 | 2.22 |
> > > | Soft-EM | 17.02 | 6.27 | 10.69 | 12.45 | 86.25 |
> > > | Soft-EM w/ Dropout Trick | 17.13 | 6.30 | 10.88 | 12.77 | 84.77 |
> > > | EqHard-EM | 22.65 | 10.13 | 27.03 | 42.02 | 22.32 |
> > >
> > > As shown, the recurrent dropout trick has minimum effect on the EM algorithm, and it does **not** prevent the non-training and synchronous-training collapses for Hard-EM and Soft-EM. This is not surprising because even Shen et al. acknowledge that their recurrent trick does not work consistently, as shown in Table 7 of their Appendix (red and blue numbers). The one-to-many phenomenon in our dialogue setting is far more severe than their machine translation setting, and it is understandable that such an ad hoc trick, without any theoretical justifications, will fail in the more difficult task.
> > >
> > > We have included the new results in the revision and updated the anonymous Git repo as well. We are still running experiments to compare with learned priors. Please let us know if you have further concerns.

---

> > > > ### Author Response · Authors · 2022-11-16
> > > > **Second followup response to reviewer ov7h**
> > > >
> > > > We have further conducted experiments to compare EM variants with uniform and learned priors.
> > > >
> > > > | Model | BLEU-1F | BLEU-2F | Dist-1 | Dist-2 | Pairwise-BLEU |
> > > > |---|---|---|---|---|---|
> > > > | Hard-EM uniform prior | 5.17 | 2.25 | 3.51 | 3.60 | 2.22 |
> > > > | Hard-EM learned prior | 4.83 | 2.32 | 9.65 | 10.02 | 0.02 |
> > > > | Soft-EM uniform prior | 17.13 | 6.30 | 10.88 | 12.77 | 84.77 |
> > > > | Soft-EM learned prior | 13.58 | 4.50 | 20.56 | 38.31 | 4.02 |
> > > > | EqHard-EM | 22.65 | 10.13 | 27.03 | 42.02 | 22.32 |
> > > >
> > > > As shown, learned priors perform worse than uniform priors due to the more severe non-training collapse problem, which is demonstrated by the extremely low BLEU scores.
> > > > This is expected because a learned prior gives the model more opportunities to select the best decoder, further exacerbating the "rich-gets-richer effect".
> > > >
> > > > Overall, these additional results justify our initial choice of not including these baselines due to their ineffectiveness and poor performance. We have nevertheless included these baselines in the revision and updated the Git repo for a more comprehensive analysis.
> > > >
> > > > We believe that we have fully addressed all your concerns about our work with these new results, and we are happy to address any further concerns or questions that you may have.
> > > >
> > > > We sincerely hope that you can read our responses and re-evaluate our work now with all the concerns and questions addressed. Thank you!

---

### Official Review · Reviewer_wvrp · 2022-10-25

**Confidence:** 4
**Correctness:** 4
**Technical Novelty And Significance:** 3
**Empirical Novelty And Significance:** 3
**Recommendation:** 6

**Clarity, Quality, Novelty And Reproducibility:**

The proposed method in the paper is novel and the ideas came from a completed analysis of related methods and the innovations beyond them.
As I said in the above weakness, the organization of the paper can be improved.

**Strength And Weaknesses:**

Strength:
The study of the literature for the solutions to justify the  one-to-many mapping phenomenon was complete. The paper also analyzed the issues coming from standard EM, Soft-EM and Hard-EM.
In the proposed new method, new neural architecture based on multi-adaptor architecture was created on top of and beyond a few state-of-the-art methods. Training methods and the theoretical analysis was given.
Weakness or questions:
It was not that clear how the Theorem 1 came up. Can the author elaborate more details of it?
The organization of the paper can be improved. It maybe better to have related work section (Section 4) come earlier because now there are duplications among Section 1 (introduction), Section 2.2 and Section 4.

**Summary Of The Paper:**

In order to solve the issue of generic responses generated by the neural dialogue systems because of one-to-many mapping phenomenon, the authors of the paper studied the state-of-the-art methods including the methods for training time and reference time, the application of the standard EM, Soft-EM, Hard EM and etc. In the paper, the analysis of the disadvantages of these methods was given and a new method EqHard-EM algorithm was proposed.
In EqHard-EM, this paper proposed a multi-adaptor architecture with multi-decoder and a shared encoder, adopted hard assignments from Hard-EM but impose equal-assignment constraints to ensure all decoders are adequately trained. To evaluate the new method, the paper included the experiments on Weibo and Open Subtitles. The results and the comparison with the state-of-the-are methods and EM variants have been given in the paper. In the results, the EqHard-EM demonstrated its advantages again EM variants and a few state-of-the-art methods.

**Summary Of The Review:**

The method in the paper is new and developed based on a good study of related method and innovations/extensions beyond them. It justifies the one-to-many mapping phenomenon and the results of the two experiments demonstrates advantages comparing to others.
This paper will benefit the readers with novel approaches.

---

> ### Author Response · Authors · 2022-11-08
> **Response to Reviewer wvrp**
>
> We thank the reviewer for recognizing our work (especially the novelty and completeness of our study) and saying that it will "benefit the readers with novel approaches".
>
> > Weakness or questions 1: "It was not that clear how the Theorem 1 came up. Can the author elaborate more details of it?"
>
> Thanks for asking. Our approach imposes an equal-assignment constraint, and Theorem 1 bounds the error when we have finite samples and an estimated posterior distribution. We proved the theorem by two lemmas (finite samples and estimated posterior, separately) and then combined them in Theorem 1. The detailed proof has been presented in Appendix A.
>
> In the revision, we provided further explanation after Theorem 1.
>
>
>
>
> > Weakness or questions 2: "It maybe better to have related work section (Section 4) come earlier because now there are duplications among Section 1 (introduction), Section 2.2 and Section 4."
>
> > Clarity, Quality, Novelty And Reproducibility: “The proposed method in the paper is novel and the ideas came from a completed analysis of related methods and the innovations beyond them. As I said in the above weakness, the organization of the paper can be improved.”
>
> Thanks again for recognizing our novelty and innovation, which were based on a complete analysis of previous literature.
>
> We have restructured the paper according to the suggestions. Specifically, we moved the related work section ahead and deleted repeated discussions about the collapses of EM training. The remaining discussion about EM in Related Work focuses on applications, and thus is not duplicated with the Introduction or Approach sections.
>
> The additional space was used for a further discussion of Theorem 1, as per your suggestion (Thanks again!).
>
> ---
>
> Summary of our response:
>
> We thank the reviewer again for the insightful review. The novelty and completeness are fully recognized by the reviewer, and the main concerns were the discussion about Theorem 1 and the location of the Related Work section, which are addressed in the revision. We’re looking forward to your stronger support!

---

> > ### Author Response · Authors · 2022-11-16
> > **Followup Response to Reviewer wvrp**
> >
> > As suggested by the reviewer, we moved the Related Work section to the second section for better clarity. We further provided a proof sketch of our theorem using the additional space in the updated manuscript as per the reviewer's request. Both are relatively minor issues.
> >
> > We thank the reviewer for the valuable suggestions and for acknowledging our work as novel and complete. We are happy to further revise our manuscript should the reviewer have more suggestions before the Period-1 deadline. We look forward to your stronger support. Thanks again!

---

### Official Review · Reviewer_EYAz · 2022-11-06

**Confidence:** 4
**Correctness:** 3
**Technical Novelty And Significance:** 3
**Empirical Novelty And Significance:** 3
**Recommendation:** 6

**Clarity, Quality, Novelty And Reproducibility:**

The paper is clearly written and easy to follow. The use of EM method for response generation which does not suffer from model collapse is novel. However, it's not clear how the approach compares to existing work that have attempted to solve similar problem of diverse response generation using GAN, reinforcement learning, or joint distribution modeling.

**Strength And Weaknesses:**

Strength:
The paper is clearly written and easy to follow. There are couple of minor typos to be fixed. The proposed method is novel and experimental results is thorough. The ablation study is also informative.

Weakness:
The use of mixture of experts for dialog modeling is not new, but the implementation using adaptor is new. The evaluation did not include existing baseline that have used GAN or reinforcement learning to accomplish diverse response generation. Also, it is not clear if encode-decoder architecture is suitable for dialog response generation as the conditional probability modeling P(y/x) exacerbate the one-to-many problem. Would be useful if authors could compare the proposed method with finetuning vanilla GPT-2 modeling P(x,y) with the evaluated dataset as explored in [1].

[1] https://arxiv.org/pdf/1908.01841.pdf

**Summary Of The Paper:**

The paper proposed the use of Equal size hard EM algorithm to address the problem of diverse response generation in open domain dialog modeling. The paper modified the EM algorithm to overcome mode collapse issues with hard EM and synchronous training collapse issues with soft EM. Experimental results show improvement in diverse response generation on Weibo and Open subtitle datasets. Ablation studies also show that Equal size hard EM performs better than beam search, soft EM and pure hard EM. Qualitative analysis also shows that some of the modes learned unique attributes of the dialog data such as response length, question mark, and like/love. The paper also shows that the performance improves with increasing number of decoders, although the incremental improvement reduces. Overall, the paper is clearly written and well motivated.

**Summary Of The Review:**

The proposed method of Equal-size hard EM is novel and well motivated and supported by experimental results. However, the paper needs to add other baselines to really evaluate the diverse response generation capability of the proposed approach.

---

> ### Author Response · Authors · 2022-11-08
> **Response to Reviewer EYAz**
>
> We thank the reviewer for saying that our paper is "clearly written and well motivated". We are especially thankful for the comment about our novelty and detailed experiments: "The proposed method is novel and experimental results is thorough. The ablation study is also informative."
>
> > Weakness: "The evaluation did not include existing baseline that have used GAN or reinforcement learning to accomplish diverse response generation"
>
> Thanks for the suggestions. We find that RL/GAN baselines are remote/less related to our approach, making it hard to have fair comparisons.
>
> Moreover, the consensus is that the RL/GAN training is too difficult to compensate for the benefit that RL/GAN could bring in. As suggested by an [EMNLP’21 paper](https://aclanthology.org/2021.emnlp-main.415.pdf) [1], “The generation performance of text GANs is not convincingly better, or even worse, than standard MLE training.”
>
> This worths significant efforts as further research, which cannot be solved by our EqHard-EM paper. In fact, we have an ongoing project that follows previous work relaxing RL training with Gumbel-softmax, but our new insight is to reduce the noise of Gumbel-softmax with sparse constraints. We’ll be happy to share that research as another paper when we have detailed results.
>
> In our revision, we point out the training difficulty of RL/GAN in Sec 1.
>
> > Weakness: "Would be useful if authors could compare the proposed method with finetuning vanilla GPT-2 modeling P(x,y)"
>
> We would like to point out that, although GPT pretraining maximizes the joint likelihood of a piece of text, its fine-tuning is usually discriminative (see the [GPT paper](https://cdn.openai.com/research-covers/language-unsupervised/language_understanding_paper.pdf) [2]), i.e., maximizing P(y|x).
>
> Even if we could model P(x,y) during training (although uncommon), the inference phase is still to obtain a response y from P(y|x). We do not see how it could solve the one-to-many problem. In fact, the one-to-many problem is the property of a task (instead of a model). We show that our mixture model, equipped with the novel EqHard-EM training algorithm, can better capture the multi-modal distribution and achieve high performance in both text quality and diversity.
> In our development, we have tried finetuning GPT-2 on the OpenSubtitles dataset, and we still have the result: 3.26 BLEU2-F, but our T5-small achieves 3.46 BLEU2-F. Therefore, we chose T5-small as the base model, because it achieves higher performance with fewer parameters. In our revised paper, we explained this in Footnote 3.
>
> [1] He, T., Zhang, J., Zhou, Z. and Glass, J., 2021. Exposure Bias versus Self-Recovery: Are Distortions Really Incremental for Autoregressive Text Generation? EMNLP 2021.
>
> [2] Radford, A., Narasimhan, K., Salimans, T. and Sutskever, I., 2018. Improving language understanding by generative pre-training. OpenAI Blog.
>
> ---
>
> Summary of our response:
>
> We thank the reviewer again for the valuable suggestions and for acknowledging the novelty and detailed experiments of our work. The main concern of the reviewer is about baselines. To this end, we provided results for GPT-2 and explained why RL/GAN is difficult to train. We also discussed them in the revision.

---

> > ### Author Response · Authors · 2022-11-16
> > **Followup Response to Reviewer EYAz**
> >
> > We thank the reviewer for the review. We have provided clarification in the author's response and revised our manuscript accordingly.
> >
> > Should the reviewer have any further suggestions, we would be happy to provide further clarification and revise our manuscript before the Period-1 deadline. Thanks again!

---

### Author Response · Authors · 2022-12-07
**Our Message to All Reviewers**

We thank all the reviewers again for their comments.

We’re expressing our concerns that, despite our month-long efforts of resolving every single question, no reviewer has updated the review or participated in the discussion, although "engag[ing] in discussion" is one of the key reviewer responsibilities set forth by the ICLR reviewer guide.

Meanwhile, we have conducted the final additional experiment about learned priors and Shen et al's dropout trick on the OST dataset (results on the Weibo dataset has already been provided in the response to Reviewer ov7h). The newest results suggest that previous methods are ineffective on OST as well, which is consistent with the Weibo dataset:

| Model | BLEU-1F | BLEU-2F | Dist-1 | Dist-2 | Pairwise-BLEU |
|---|---|---|---|---|---|
| Hard-EM uniform prior | 9.20 | 1.56 | 9.87 | 10.60 | 0.61 |
| Hard-EM learned prior | 4.70 | 0.70 | 8.60 | 6.57 | 0.00 |
| Soft-EM uniform prior | 15.52 | 3.00 | 13.23 | 15.93 | 70.29 |
| Soft-EM learned prior | 15.47 | 2.95 | 13.40 | 16.17 | 67.46 |
| EqHard-EM (w/o dropout trick) | 17.86 | 3.76 | 34.09 | 50.41 | 13.15 |

As there is less than one week before the deadline of period-2 discussion, we urge all reviewers to check our responses and revised paper, and update their reviews accordingly. We authors will take on-call shifts every day to ensure that any further questions can be answered within 24 hours.

Thank you!

---

### Decision · Program_Chairs · 2023-01-20

**Decision:**

Accept: poster

**Justification For Why Not Higher Score:**

The paper is clear in general, but can be presented in a more self-contained way, with references to issues, such as decoder collapse problem.

**Justification For Why Not Lower Score:**

Reviews include constructive suggestions and misunderstandings, which were clarified by the authors. The proposed approach is interesting and useful, and tackles an important problem, hence the paper is expected to attract interest from the response generation community.

**Metareview: Summary, Strengths And Weaknesses:**

This paper tackles the well-known generic response problem of neural response generation systems, due to the one-to-many issue. It provides a good summary and categorization of related work for generating more diverse responses and proposes a modified EM algorithm, EqHard-EM algorithm, to train a multi-decoder model for improved response diversity. The model used in this work includes multiple decoders with adaptors and a shared encoder. EqHard-EM assigns each decoder a sample and imposes an equal assignment constraint to ensure all decoders are well-trained.  Experimentation on Weibo and Open Subtitles datasets compares the proposed approach with state-of-the-art methods and EM variants (Soft-EM, Hard-EM, etc.), demonstrating the advantages of the proposed approach.
Reviewers made a set of suggestions, including experimentation with GPT2, including results with BLEU variants, and including detailed run-time analysis, which are added during the author responses. There seem to be some misunderstandings on the first version of the paper, which the authors clarified in their response.

**Note From Pc:**

if the above contains the word "oral" or "spotlight" please see: "oral" presentation means -> notable-top-5% and "spotlight" means -> notable-top-25%. As stated in our emails, we are disassociating presentation type from AC recommendations